# WALES 2021 Active Healthy Kids (AHK) Report Card: The Fourth Pandemic of Childhood Inactivity

**DOI:** 10.3390/ijerph19138138

**Published:** 2022-07-02

**Authors:** Amie B. Richards, Kelly A. Mackintosh, Nils Swindell, Malcolm Ward, Emily Marchant, Michaela James, Lowri C. Edwards, Richard Tyler, Dylan Blain, Nalda Wainwright, Sarah Nicholls, Marianne Mannello, Kelly Morgan, Tim Evans, Gareth Stratton

**Affiliations:** 1Applied Sports, Technology, Exercise and Medicine (A-STEM) Research Centre, Swansea University, Swansea SA1 8EN, UK; k.mackintosh@swansea.ac.uk (K.A.M.); n.j.swindell@swansea.ac.uk (N.S.); malcolmward@ntlworld.com (M.W.); g.stratton@swansea.ac.uk (G.S.); 2Population Data Science, Medical School, Singleton Campus, Swansea University, Swansea SA2 8PP, UK; e.k.marchant@swansea.ac.uk (E.M.); m.l.james@swansea.ac.uk (M.J.); 3Cardiff School of Sport and Health Sciences, Cardiff Metropolitan University, Cardiff CF23 6XD, UK; lcedwards@cardiffmet.ac.uk; 4Health Research Institute and Movement Behaviours, Nutrition, Health, Development, and Wellbeing Research Group, Department of Sport and Physical Activity, Edge Hill University, Ormskirk L39 4QP, UK; tylerr@edgehill.ac.uk; 5Institute of Management and Health, Carmarthen Campus, University of Wales Trinity Saint David, Carmarthen SA31 3EP, UK; d.blain@uwtsd.ac.uk (D.B.); n.wainwright@uwtsd.ac.uk (N.W.); 6Department of Business, School of Management, Bay Campus, Swansea University, Swansea SA1 8EN, UK; sarah.nicholls@swansea.ac.uk; 7Play Wales, Park House, Greyfriars Road, Cardiff CF10 3AF, UK; marianne@playwales.org.uk; 8DECIPHer, School of Social Sciences, Cardiff University, SPARK, Cardiff CF24 4HQ, UK; morgank22@cardiff.ac.uk; 9Sport Wales, Sport Wales National Centre, Sophia Gardens, Cardiff CF11 9SW, UK; tim.evans@sport.wales

**Keywords:** health, sedentary, play, policy, community and environment, school, children, physical activity

## Abstract

This is the fourth Active Healthy Kids (AHK) Wales Report Card. The 2021 card produced grades on children and young people’s physical activity (PA) using pre-COVID-19 data that were not used in previous versions. Eleven quality indicators of PA were graded through expert consensus and synthesis of the best available evidence. Grades were assigned as follows: Overall PA—F; Organised Sport and PA—C; Active Play—C+; Active Transportation—C−; Sedentary Behaviours—F; Physical Fitness—C−; Family and Peer Influences—D+; School—B−; Community and the Built Environment—C; National Government and Policy—C; and Physical Literacy—C−. All but three grades remained the same or decreased from the 2018 AHK-Wales Report Card (Active Play increased from C− to C+; Active Transportation, D+ to C−; Family and Peers, D to D+). This is concerning for children’s health and well-being in Wales, particularly given recent evidence that PA has further decreased during the COVID-19 pandemic. The results from the Report Card should be used to inform the decision making of policy makers, practitioners and educators to improve children and young people’s PA levels and opportunities and decrease PA inequalities.

## 1. Introduction

Children’s physical activity (PA) levels worldwide are poor and have been decreasing over the decades [1] to the detriment of physical [2] and mental health [3]. This is, unfortunately, no different in Wales (United Kingdom; U.K.) [4]. Wales has a population of approximately 3 million people, of which approximately 664,000 are children and young people aged between 0 and 18 years [5]. Only 51% of the Welsh population who are aged 3–17 years meet the guidelines for PA. These guidelines recommend at least 60 minutes of PA every day of the week [6]. Only 13–17% of children aged 11–16 years partake in the recommended amount of PA [7], leaving Wales with some of the poorest levels of PA and time spent in sedentary behaviour globally [8,9].

Guidelines suggest that PA should be of moderate to vigorous intensity and that some PA is better than none. The guidelines further recommend that children and young people engage in a variety of types and intensities of PA to develop movement skills, muscular fitness and bone strength, and that sedentary time should be minimised [6]. Poor levels of PA have a subsequent negative effect on physical health, including increased risk of obesity [10]. In Wales, this is reflected in data from the Public Health Wales Child Measurement Programme [11], which show year-on-year weight increases with more than one in four children (aged 4–5 years) now living with overweight or obesity in Wales. High levels of obesity are also linked with low levels of fitness [12]. Research in Wales indicated that less than half of 3000 9- to 11-year-old children assessed achieved a healthy level of overall health-related fitness [13]. Furthermore, research has shown that PA levels during childhood can be tracked into adulthood, with children who are physically active more likely to be physically active as adults [14]. Thus, there is a growing need to (i) highlight the current PA situation with children and young people across Wales; (ii) identify any inequalities in achieving recommended levels of PA in Wales; and (iii) provide recommendations for improving these to inform the decision making of policy makers, practitioners and educators in Wales. These needs have been recognised by policy makers who previously produced a report detailing 20 recommendations to improve children and young people’s PA in Wales [7].

There are many inequalities in PA levels, including, but not limited to, age, sex, race and ethnicity and socioeconomic status. Specifically, the current evidence base suggests that boys are more active than girls [15]; PA levels decline with increasing age [16]; differences exist between ethnic groups, with White European children recording more PA than South Asian children [17,18]; and lower socioeconomic status is associated with lower PA levels [19]. Despite these inequalities, Wales has a strong performance history of supporting PA through policies and strategies, with the 2018 Active Healthy Kids (AHK) Wales Report Card awarding a grade of C+ based on the identification of 21 national documents with a direct bearing on children and young people’s PA. However, the Report Card did highlight that some of these policies were outdated [4,20].

The Active Healthy Kids Global Alliance (AHKGA) is a network of researchers and health professionals aiming to improve children and young people’s PA across the world, with an expanding development over time [21]. The 2021 Report Card for Wales is the fourth Active Healthy Kids Wales (AHK-Wales) Report Card, following reports in 2014, 2016 and 2018. Wales’s fourth Report Card contributes to the AHKGA Global Matrix 4.0, which consists of 57 countries across six continents, after previously producing Report Cards for the Global Matrix 1.0 [22], 2.0 [23,24] and 3.0 [4,9]. The AHK-Wales Research Working Group (RWG) is co-ordinated by Swansea University and consists of academics and professionals from across Wales in the fields of sport, play, transport, public health, leisure and education. The overall purpose of the AHK-Wales Report Card is to act as an advocacy tool and provide a lens over PA behaviours and influencing factors of children and young people within Wales. A strength of the AHKGA process is that participating countries use a standard method that allows for comparison between countries, and it highlights children and young people’s rights to be active and healthy. The purpose of this paper is to summarise the findings of the AHK-Wales 2021 Report Card and to present and interpret the results for PA domains for children and young people.

## 2. Materials and Methods

The AHK-Wales RWG consisted of 20 members, including academics, postgraduate researchers, professionals and practitioners with expertise in PA and access to national data sources. The Academic Lead (G.S.) gained funding for and supervised the project, whilst the Lead Researcher (A.B.R.) led on sourcing and synthesising the data, organising the quality indicator (QI) groups and liaising with the AHKGA regarding progress of the Wales RWG. G.S. and A.B.R. were also responsible for producing the Report Card and subsequent impact activity, including that of the website, social media and further advocacy practices. Members of the RWG were allocated to QI groups, each with an associated lead (initials below), to collectively allocate grades to each QI.

The method used throughout the process was aligned to guidance from the AHKGA [22] culminating in the production of eleven PA QI grades. Whilst Physical Literacy was not an AHKGA QI, the RWG deemed the inclusion important, given the association with overall PA [25] and the ongoing work on this in Wales over recent years [26,27]. The eleven PA QIs were assessed and graded using the AHKGA standardised Report Card development process, which has been previously described for the Global Matrix 2.0 [23] and 3.0 [9] and is based on the Canadian Report Card model [28]. Briefly, this process involved collating, synthesising and providing expert consensus on the best available data aligned to the AHKGA benchmarks for the Global Matrix 4.0 (Table 1) for each QI. Slight alterations of benchmarks were made between Global Matrix 3.0 and 4.0 based on feedback from stakeholders. The RWG decided to use pre-COVID-19 data to produce the 2021 Report Card due to the time gap between the 2018 Report Card and the 2021 Report Card. However, it should be noted that a further Report Card is currently in production to compare the grades in this Report Card to children and young people’s PA post-COVID-19. The eleven QIs included Overall Physical Activity (K.A.M.), Organised Sport and Physical Activity (R.T.), Active Play (M.M.), Active Transportation (N.S.), Sedentary Behaviours (K.M.), Physical Fitness (D.B.), Family and Peers (S.N.), School (E.M.), Community and Environment (M.J.), Government (M.W.) and Physical Literacy (N.W.). The QIs were graded using a standardised grading scheme and rubric (Table 2), provided by AHKGA, ranging from A+ (94–100% of children met the criteria) to F (<20% met the criteria) or Inconclusive (Inc), where data were inadequate or not available. The most recent and highest quality data (at the time of the grading process pre-COVID-19) were used from the following data sources: Health and Attainment of Pupils in a Primary Education Network (HAPPEN), School Health Research Network (SHRN), School Sport Survey, Further Education Sport and Active Lifestyles Survey, Play Sufficiency Child Survey Analysis, The National Survey for Wales, Swan-Linx and Bridge-Linx Data, Dragon Challenge Data, and Movement Assessment Battery for Children (MABC) Data (Table 3). Each of these data sources had various sample sizes and age ranges (Table 3) and these were taken into consideration when grading the QIs. Information about the accessibility of these data sources can be found in Appendix A, Table A1. Data sources that were nationally representative were prioritised, followed by the best available data for each specific QI. The RWG regularly met virtually using an online platform to discuss and critique data for each of the eleven Qis, resulting in a grade being presented. During the meetings, data were reviewed in terms of representativeness, data collection, age range of participants, sample size and the reporting of any inequalities, including age, sex, race and ethnicity and socioeconomic status.

## 3. Results

The overall grades for each QI are presented in Table 4. Results are discussed herein on an individual QI basis.

### 3.1. Overall Physical Activity Levels: F

In line with the global PA guidelines [5], “the percentage of children and young people who accumulate at least 60 minutes of moderate-to-vigorous PA (MVPA) on all day seven days of the week” was used as the benchmark for this QI. Data from the SHRN and HAPPEN databases were used to provide a grade (Table 3). The SHRN findings indicated that 17% of young people were active for at least 60 minutes on all seven days, although this figure decreased to 14% when only considering MVPA. There were inequalities in these findings, with more boys (18%) reporting engaging in MVPA in comparison to girls (10%). An age gradient was also present, with younger children (11-year-olds) reporting higher levels of MVPA (20%) in comparison to 16-year-olds (10%). A lower percentage (12%) of young people from less affluent families met the PA guidelines, compared to more affluent families (17%); different ethnicities reported similar PA levels (White: 14%; Black Minority Ethnic: 15%). The HAPPEN survey provided insight into the PA levels of 8–11-year-old children, with 22% reporting carrying out sport/exercise, as the structured component of PA, for at least 60 minutes across all seven days. Moreover, there were similar inequalities whereby boys reported more engagement (27%) in sport/exercise than girls (17%). There was, however, little variation with age (21–24%) or socioeconomic status (20–24%). Taken together, the RWG assigned a grade F to this QI. Although this suggests a decrease from the 2018 AHK-Wales Report Card, it is pertinent to note that the benchmark and data used in this Report Card differed from previous report cards, thereby precluding direct cross-comparisons. Indeed, some MVPA data were used as well as considering PA levels across all seven days, rather than “on average” or “on at least four days”, commensurate with the AHK Global Matrix 4.0 benchmarks. Nonetheless, the benchmarks align closer to Wales’s national PA guidelines.

### 3.2. Organised Sport and Physical Activity: C

The benchmark for grading this indicator was aligned with Sport Wales’s concept of children and young people being “hooked on sport”, meaning taking part in sport on three or more occasions per week. Data were captured in the Sport Wales School Sport Survey and are consistent with benchmarks used in previous AHK-Wales Report Cards (2014 and 2016). The School Sport Survey and the Further Education Sport and Active Lifestyles Survey showed the following figures for taking part in sport or PA on three or more occasions per week: 44% (7–9 year olds, school years 3 and 4); 51% (9–11 year olds, school years 5 and 6); 49% (11–14 year olds, school years 7–9); 46% (14–16 year olds, school years 10 and 11); 44% (16-year-olds); 46% (17-year-olds). This equates to an average of 47% and thus a C grade. Sub-group analysis (age, sex, ethnicity, disability and socioeconomic status) revealed that inequalities exist. There remains a sex difference in participation levels, with 50% of boys and 46% of girls hooked on sport. Furthermore, socioeconomic inequalities have widened since the previous Report Card, with the gap in participation rates between the least- and most-deprived children increasing by two percentage points. In both primary and secondary school data, the gaps have closed between ethnicities/ethic groups and between those who are impaired/disabled and those who are not since the previous Report Card. Despite this, disparities between ethnic groups and disabilities do exist within further education, with children from a Black/African/Caribbean/Black British ethnic group being more likely to participate in sport/PA three or more times a week (45%) than White (43%) and Asian/Asian British (35%) ethnic groups. Furthermore, there remains a difference between those who are impaired/disabled and those who are not, with 33% of students with an impairment/disability participating three or more times a week in 2018, compared with 45% of those who did not identify an impairment/disability. This gap has unfortunately widened since the last Report Card. The Organised Sport and Physical Activity grade remains unchanged from the 2016 AHK-Wales Report Card, but has decreased from the 2018 AHK-Wales Report Card. It is important to note that because of varying survey cycle dates, different questions and surveys were used for the grading of the 2018 Organised Sport Participation indicator compared to the 2014, 2016 and 2021 Report Cards. Therefore, caution is advised when viewing the changes in grades across Report Cards.

### 3.3. Active Play: C+ 

For the benchmark “percentage of children and young people who report being outdoors for several hours a day”, information from the Play Sufficiency Child Survey 2018–2019 was used. The survey asked children how often they go out to play or hang out with friends. When asked if they played outside most days, 42% of children aged 5–17 years reported that they did and 33% of children reported playing outside a few days each week. For the benchmark “percentage of children and young people who engage in unstructured/unorganised active play for several hours a day”, the SHRN Student Health and Wellbeing Survey 2019/20 was used, as playing with friends is included in the definition of exercise within the survey. In this survey, 42% responded “often or more” to the question “how often, during the most recent summer holidays, did you exercise in your free time so much that you got out of breath or sweated?” When taking the Play Sufficiency Child Survey Analysis responses to the question “days spent playing out” into account, the RWG slightly deviated from the recommended >2 hours per day. The RWG determined that responses of “most days” and “a few days” each week were representative of the benchmark “several days” a week. Taking into consideration both benchmarks, the RWG concluded that Active Play within Wales had an overall percentage of 58%, which is commensurate with a C+ grade. This grade is slightly improved from the C− grade previously reported in 2018. However, it is important to acknowledge that only participants who indicated the response “most days” each week were included in the data for the 2018 Report Card.

### 3.4. Active Transportation: C−

This RWG determined a grade of C− for this indicator based on the benchmark, “percentage of children and young people who use active transportation to get to and from places (e.g., school, park, mall, friend’s house)”. According to three surveys (SHRN Student Health and Wellbeing, HAPPEN and Play Sufficiency Child Survey Analysis), between 42% and 44% of primary school-aged children use active transport to travel to school. In two surveys of children aged 11–16-years, 35% and 33% used active transport to travel to school. In another survey, 73% of children aged 4–18 years used active travel to places where they play. A small sex difference was shown, with a higher proportion of boys using active transport to primary school compared to girls, whereas, at secondary school level, this trend was reversed, with more girls using active transport compared to boys. The proportion of children using active transport was also greater in urban compared to rural primary and secondary schools. The proportion of children using active transport also increased with age, with 45% and 51% of children in Year 6 using active travel to and from school, respectively, compared to 37% and 41% in Year 4. When drawing comparisons across socioeconomic groups, there was a clear gradient, with the most affluent group reporting higher active travel (37%) compared to the most deprived (21%). The Play Sufficiency Child Survey Analysis showed that 73% of children used active transport to travel to places of play. The RWG assigned a C− to this category, considering that active transport to school ranged between 33% and 43% and accounting for the first inclusion of a question addressing active transport to a destination other than school. This grade has increased from a D+ in the last AHK-Wales Report Card completed in 2018. It is important to note that the grade increase reflects the inclusion of more data sources as well as the inclusion of data on active transport to destinations other than school. Thus, this grade provides a more complete overview of this indicator, rather than reflecting evidence of an upward trend in the use of active transport. 

### 3.5. Sedentary Behaviours: F

There is one benchmark for the Sedentary Behaviours indicator, which is “the percentage of young people who exceed the recommended sedentary time guidelines (i.e., two or more hours)”. Two data sources were used to assign a sedentary behaviour grade, namely, the SHRN, which asked young people how much time they spent sitting outside of school hours during free time on weekdays and the HAPPEN survey, which asked children “in the last seven days, how many days did you watch TV/play online games/use the internet etc. for two more hours a day (in total)?” These data showed that 86.4% of young people spent two or more hours sitting during weekdays. Results were found to be higher amongst boys and there was a positive association between sedentary behaviour and age. Data also showed that 32% of children reported watching TV/screens for two hours or more every day of the week, which was also higher in boys than girls, and in older children. There was also a difference in socioeconomic status, where 37% of deprived and 21% of the most affluent children reported two or more hours of screen time daily. A grade F was reported for this indicator, which has not changed since the 2018 Report Card.

### 3.6. Physical Fitness: C−

The overall grade for physical fitness was decided upon using the best available data to compare to the Tomkinson et al. (2018) normative values as per the AHKGA benchmark, “average percentile achieved on certain physical fitness indicators based on the normative values published by Tomkinson et al. (2018) [30]”. Available data for both cardiorespiratory fitness and muscular fitness were used. Based on the best available data from the Swan-Linx and Bridge-Linx Projects, the RWG assigned a C− to this indicator. For cardiorespiratory fitness, data from the multi-stage fitness test for 4310 participants aged 9–12 years were available. Overall, the mean lap achieved was 29.65 laps (SD = 16.76). For comparison with European normative values [30], data were classified by age and sex, with children and young people in Wales typically falling within the 40th percentile for cardiorespiratory fitness. Results for muscular fitness also showed children and young people falling into the 40th percentile, using data from the grip strength test for 3387 participants aged 9–12 years comparing with European normative values [30], classified by age and sex. Overall, young people in Wales aged 9–12 typically fall within the 40th percentile for both cardiorespiratory and muscular fitness, meaning that the physical fitness indicator was graded a C−.

### 3.7. Family and Peers: D+

The RWG allocated the Family and Peers indicator the grade of a D+ based on data from three out of the five AHKGA recommended benchmarks (Table 1). Data from the Play Sufficiency Child Survey Analysis and the National Survey for Wales were used to evaluate these benchmarks. Children reported that 46% of adults were “great and happy with children playing out”, with another 43% stating that adults were “OK and alright about children playing out”. Data show that 10% of adults volunteered in sport over the past 12 months, whilst 53% of adults met the MVPA guidelines of 150 minutes of PA per week. Based on these results, an average was taken, giving a total of 36%, equating to a grade D+, a slight increase from the 2018 Report Card, which was a D.

### 3.8. School: B−

To grade this indicator, data from the HAPPEN Survey, SHRN School Environment Questionnaire, SHRN Health and Wellbeing Survey and School Sport Survey Participation and Provision surveys were synthesised to provide data on the six benchmarks provided for by the AHKGA (Table 1). The data used to produce a grade for the school indicator were as follows: 45% of primary schools offered an afternoon break. When exploring PE specialisations within teaching, 69% of primary schools and 92% of secondary schools had at least one male specialist PE teacher, whilst 69% of primary and 80% of secondary schools had at least one female specialist PE teacher. When exploring how many minutes per week were allocated to PE within the formal curriculum, only 6% of children aged 11–12 years (school year 7) were offered the recommended 120 minutes per week, which decreased with age, with <1% of children aged 15–16 years offered this amount. When exploring extracurricular sport, 84% of primary (7+ years old) and 94% of secondary schools were offered regular access to extracurricular sport, whilst on average, 66% across primary and secondary schools reported participation (68% males; 65% females). Finally, when studying resources, equipment and facilities, it was found that indoor and outdoor space conducive to facilitate PA were provided to 72% of schools, whilst 64% of primary and 58% of secondary staff agreed/strongly agreed that their school has access to sufficient facilities to provide sport. The RWG recognises these benchmarks to be of equal importance and assigned an even weight, allocating an overall average of 60% and a grade of B−.

### 3.9. Community and Environment: C

The RWG decided on an overall grade of C for the community and built environment indicator. The main data sources considered for this Report Card included a broader range of data. The National Survey for Wales, HAPPEN survey and Play Sufficiency Child Survey Analysis provided the data for the three benchmarks evaluated within this indicator. Children and parents who perceive that their community/municipality is doing a good job at promoting PA (e.g., variety, location, cost, quality) were evaluated through several questions: 21% report they were satisfied with places to play; 23% were very satisfied with clubs and activities; 21% were very satisfied with places to meet; 48% strongly agree that green space was suitable; 88% of children were happy with their area. These questions averaged out at 40%, equating to a C−. Children and parents reporting that they have facilities, programs, parks and playgrounds available to them in their community varied between the National Survey for Wales and HAPPEN survey. A total of 72% of children reported playing out, 24% reported being able to play everywhere they would like, 42% reported great places to play, whilst 88% can walk to a park and 38% can walk to a facility, averaging this benchmark at 53% (C). When exploring the percentage of children or parents who report living in a safe neighbourhood where they can be physically active, 45% reported feeling safe in the Play Wales Survey, whilst 70% reported feeling safe in the HAPPEN survey. An average of 58% (C) was calculated for this benchmark. Overall, those who were more deprived reported feeling less safe in their areas. Additional data sources have been made available since 2018 and it is apparent that the evidence base is growing in this area; however, it is still reliant on self-report data from both children and adults. The grade of C has been assigned to this indicator based on 3/6 of the benchmarks, highlighted in Table 1. For previous Report Cards, the RWG used the percentage of children/parents satisfied with the play facilities available in their local area to assign a grade to this indicator. Data available around these benchmarks were limited. 

### 3.10. Government: C

The RWG concluded that the grade would be decreased slightly from a C+ in 2018 to a C (50%) for this current Report Card. This is in part due to the expiry of previous policies specific to PA promotion that were subsequently replaced by an obesity policy that includes PA as one element. This was seen by the group as a retrograde step as it overlooks the wider health impacts of PA. There was also a perceived lack of progress in embedding PA in education policy and actions despite a recent revision of the national curriculum in Wales. Similarly to the other QIs, there are no purely objective measures that can be used to inform the Report Card. However, after utilising the WHO Europe Health-enhancing PA (HEPA) policy audit tool (PAT)v2 [31] to inform the 2018 Report Card, we developed a complementary weighted scoring tool that provided an objective measure aligned to the Report Card [20]. In interpreting this indicator, twenty-six “active” national instruments including national policies, strategies, action plans, legislation and a few other advisory and technical documents that have a direct bearing on children and young people’s PA were identified. Guided by the HEPA PAT tool, the evidence relating to key policy domains that influence PA in children and young people were evaluated. A range of key “elements” were identified from the HEPA PAT tool and refined by the RWG that could individually or collectively impact the effectiveness of the policy instrument. These elements included number and breadth of policies, identified supporting actions, identified accountable organisation(s), identifiable reporting structures, monitoring and evaluation plans and identified funding/resourcing. The HEPA PAT tool was used in assessing this indicator and the information translated into a score by utilising the previously used scoring tool [20]. Each element was assigned a percentage score weighted to reflect the element’s perceived importance in translating the policy instruments effectively. The final scoring matrix was as follows: number and breadth of relevant policies—10% (5% number and 5% breadth) = 8%, identified supporting actions—20% = 15%, identified accountable organisation—25% = 14%, identifiable reporting structures—15% = 6%, identified funding and resources—20% (5% number of identified national programmes and 15% funding) = 5%, monitoring and evaluation plan—10% = 2%.

### 3.11. Physical Literacy: C−

Following the inconclusive grades assigned to physical literacy in 2016 and 2018, a C− grade was awarded. The definition of physical literacy utilised in Wales is “the motivation, confidence, physical competence, knowledge and understanding to value and take responsibility for engagement in physical activities for life” [32,33]. Following the elements of physical literacy in this definition, the concept was divided into four sub-indicators: physical competence, motivation, confidence and PA as a behaviour that is representative of physical literacy. There were no available data for the cognitive (knowledge and understanding) domain; however, there were data from six sources to support a score in physical competence, motivation, confidence and PA. Using the six sources highlighted in Table 3 (Dragon Challenge, Movement ABC, HAPPEN, School Sport Survey, FE Sport and Active Lifestyle Survey, and SHRN), each domain was scored as follows: physical competence, 34%; confidence, 69%; motivation, 65%; PA, 19%. Based on these scores for the sub-categories, the overall score is 47%, giving a grade of C−.

## 4. Discussion

The results of the AHK-Wales 2021 Report Card have shown that prior to COVID-19, children and young people’s PA levels in Wales decreased. Of concern, recent research during and after the COVID-19 pandemic shows that on average, these negative trends further declined during pandemic restrictions [34,35]. Using the best available data, the majority of children in Wales were insufficiently active, did not achieve the recommendation of at least 60 minutes of MVPA every day and spent excess time (>2 hours) in sedentary behaviour [36]. This was the first year the Wales RWG was able to report a grade for all eleven QIs. All but three grades (Active Play, Active Transportation and Family and Peers) either remained the same or decreased from 2018 to 2021. This is of great concern, not least given that recent research has also shown that pandemic and post-pandemic PA levels in Wales have further decreased [34,37]. Based on this recent research and the results of the Report Card, a further Report Card detailing children and young people’s PA grades post-COVID-19 will be released shortly.

Quality indicators of particular concern include Overall Physical Activity and Sedentary Behaviours, both receiving a grade F, the lowest of the Wales indicators and decreasing from previous years. The Physical Literacy and Physical Fitness QIs were given grades for the first time in the AHK-Wales Report Card history. Given the importance of both physical fitness [38] and physical literacy [39] to overall health and well-being, considerations are needed on how data can be maintained to ensure effective monitoring of these QIs for the next Report Card cycle and beyond. The recent global pandemic has only further highlighted the importance of good physical fitness [40]. The School QI was given a grade of B−, which has decreased from the B grade assigned in the 2016 Report Card. Data behind this grade suggest that the provision and resources are available; however, time allocated to PE within schools is not sufficient. The importance of the provision of facilities and the duration of both PE lessons and lunchbreaks in facilitating PA and reducing sedentary behaviour has been previously established [41]. The results from the School QI are particularly relevant to the new curriculum in Wales coming into effect in 2022. Despite a renewed statutory focus on *Health and Wellbeing* as one of the six *Areas of Learning and Experience*, there will not be mandated time requirements for PE provision. Both Community and Environment, and Family and Peers QIs were able to assign grades; however, data were not available for all suggested benchmarks, and hence the grades were assigned based on an appropriate selection of the benchmarks, as shown in Table 1. This should be taken into consideration when evaluating these QIs.

Congruent with all three previous Report Cards [4,24], there were inequalities in children and young people’s PA across Wales. These inequalities are at risk of widening, especially following the COVID-19 pandemic, particularly the sex/deprivation gap. Whilst there was a significant increase in both walking and cycling, with 38% and 39% increases, respectively, compared to pre-COVID levels [42], these increases in activity were identified amongst people predominantly at the higher end of the socioeconomic spectrum. However, other studies suggest a reduction in PA amongst more vulnerable and disadvantaged groups [43], which would subsequently further widen the health inequalities gap. 

### 4.1. Research Gaps

The AHK-Wales Report Card is typically produced every two years, but the COVID-19 pandemic resulted in a three-year gap between this Report Card and the previous one in 2018. Following in-depth discussions, the RWG came to a consensus to focus this Report Card on pre-COVID-19 data. During the production of the 2014, 2016 and 2018 Report Cards, there were significant gaps identified within the data, resulting in INC grades (n = 4 in 2018) for Physical Fitness, Physical Literacy, School and Community and Environment. Previous research has identified both national and international research gaps and no ideal PA surveillance measures [44]. It is therefore particularly noteworthy that all indicators were assigned grades in the 2021 Report Card due to increased availability of data. Within the Physical Fitness QI, data from the Swan Linx and Bridge Linx programmes were used to establish levels of fitness in children. However, whilst there was a large sample size (n = 4778), these data are not nationally representative and thus there remains a paucity of national-level data. Despite this, the deprivation levels, measured through the Welsh Index of Deprivation (WIMD), align with and span the socioeconomic status of Wales as a nation. Similarly with the Physical Literacy QI, although substantially more data are available in this area, particularly from the School Sport Survey and the Dragon Challenge, which are both nationally representative sources and provide data for physical competence, confidence, motivation and PA, there were no data available for the knowledge and understanding aspect of physical literacy. Furthermore, there is a continued lack of data that consider the holistic approach to physical literacy. The School QI was primarily graded INC in the 2018 Report Card as a result of the Sport Wales School Sport Survey being conducted every three years instead of every two, producing no updated data since the 2016 Report Card. Finally, the Community and Built Environment QI, which only had sufficient data for three out of the six benchmarks suggested by the AHKGA, was able to be graded, in accord with other countries, following consultation and audit from the AHKGA. Nonetheless, efforts are required to provide data for all benchmarks and enable a more comprehensive overview of the indicator. 

Despite there being an improvement in the range of data available for the 2021 Report Card, there are still no large-scale nationally representative studies utilising device-based PA or sedentary time measures, which would be beneficial over solely self-reported information. Furthermore, there remains a scarcity of qualitative evidence to provide detail on the quality of PA provision, or indeed ascertain why PA levels are changing. To address why grades have been awarded, the voices of children could be utilised to apply context. While these data are difficult to obtain on a larger scale, it is important to advocate for what children want and need to be more active. Finally, another issue that should be considered is how closely linked the available data were to the AHKGA proposed benchmarks. Indeed, some data were not as closely aligned to the QIs as others, with some indictors requiring deeper exploration, more synthesis and the use of expert opinion. It is important to note that this Report Card has also taken a more robust approach from more experienced RWG members who have been involved in the process for four iterations. 

### 4.2. Recommendations to Improve the Grades

This is the fourth AHK-Wales Report Card to be published and the first to include a grade for all QIs. As the RWG has developed and become more established, the lasting impact, reach and significance of the Report Card increase with importance. The results can be used to inform the decision making of policy makers, practitioners and educators. Indeed, there are several key recommendations that will be advocated for based on these results. Specifically, the Active Play QI suggests prioritising the views of children and young people, making the most of community assets and protecting play time, whereas the Physical Fitness QI suggests that health enhancing PA should be encouraged from a range of contexts. Moreover, based on the Community and Environment QI, the inclusion of more data sources with children’s voices—as data sources are currently predominantly adult-based—should be prioritised, along with the development of objective measures of access, including, but not limited to, network distances. However, it is pertinent to note that such objective measures should be used in conjunction with self-report so that subjective accessibility is also monitored. Finally, the key recommendation from the Government QI is that PA should not be conflated with “obesity”; whilst there is a relationship, the health impacts of PA extend far beyond obesity. Health-enhancing PA is influenced by most policy areas, and therefore, any strategic approach must ensure active engagement with all relevant sectors, including education, communities, environment, play, transport, health and sport. 

A primary concern following these grades is the evident decline in children and young people’s PA levels in Wales. It is therefore imperative to seek to positively impact children’s PA levels between the Report Card cycles. Data from the 2018 AHK-Wales Report Card were used in the National Assembly for Wales Health, Social Care and Sport Committee Report on PA of children and young people in March 2019 [7]. Since the last Report Card, The Welsh Institute of Physical Activity, Health and Sport (WIPAHS) has been inaugurated. WIPAHS brings together academia, facilitators, policy makers and the public to develop and answer questions on the nation’s well-being and health, whilst translating this research into practice. It is anticipated that WIPAHS will play a key role in sharing data and research to further inform the AHK-Wales RWG.

### 4.3. Future Directions

The dissemination of the Report Card will continue with the launch of the Report Card at the AHKGA meeting in conjunction with the International Society for PA and Health (ISPAH) in October 2022; this will provide opportunities to learn from other countries that have followed the same process. This will be supported by an independent website (www.activehealthykidswales.net accessed on 24 June 2022). These are the foundations for continued work with the advocacy and impact group, who will resume enhancing communications with key stakeholders to improve the PA of children and young people in Wales.

## 5. Conclusions

Worryingly, in comparison to the AHK-Wales 2018 Report Card, eight out of the eleven QIs remained the same or decreased in the 2021 Report Card, with the largest decrease being in the Overall Physical Activity QI, which changed from a D+ to an F. The AHK-Wales 2021 Report Card emphasises the hard work of academics and professionals in Wales in developing, implementing and analysing new datasets since 2018, which has allowed more data to be used for the 2021 Report Card. This has allowed a more comprehensive understanding, and in some instances, has helped provide further context. However, it is pertinent to acknowledge that the current AHKGA benchmarks are open to interpretation, and some measures utilised were not fully aligned with Global Matrix 4.0 benchmarks. It is therefore recommended that national surveillance strategies are implemented to provide a more holistic and informative overview of children and young people’s PA.

## Figures and Tables

**Table 1 ijerph-19-08138-t001:** The Active Health Kids Global Alliance benchmarks for each quality indicator. The benchmark for Physical Literacy was created by the Active Healthy Kids-Wales group.

Indicator	Benchmark(s)
Overall Physical Activity	% of children and young people who meet the Global Recommendations on Physical Activity for Health [6], which recommend that children and young people accumulate at least 60 minutes of moderate-to-vigorous-intensity physical activity per day on average.* OR % of children and young people meeting the guidelines on at least four days a week (when an average cannot be estimated).
Organised Sport and Physical Activity	% of children and young people who participate in organised sport and/or physical activity programs.*
Active Play	% of children and young people who engage in unstructured/unorganised active play at any intensity for more than two hours a day.* % of children and young people who report being outdoors for more than two hours a day *.
Active Transportation	% of children and young people who use active transportation to get to and from places (e.g., school, park, mall, friend’s house) *.
Sedentary Behaviours	% of children and young people who meet the Canadian Sedentary Behaviour Guidelines [29] (5- to 17-year-olds: no more than two hours of recreational screen time per day). Note: the Guidelines currently provide a time limit recommendation for screen-related pursuits, but not for non-screen-related pursuits *.
Physical Fitness	Average percentile achieved on certain physical fitness indicators based on the normative values published by Tomkinson et al. [21] *
Family and Peers	% of family members (e.g., parents, guardians) who facilitate physical activity and sport opportunities for their children (e.g., volunteering, coaching, driving, paying for membership fees and equipment) *. % of parents who meet the Global Recommendations on Physical Activity for Health [5], which recommend that adults accumulate at least 150 min of moderate-intensity aerobic physical activity throughout the week or at least 75 min of vigorous-intensity aerobic physical activity throughout the week, or an equivalent combination of moderate- and vigorous-intensity physical activity *. % of family members (e.g., parents, guardians) who are physically active with their kids. % of children and young people with friends and peers who encourage and support them to be physically active *. % of children and young people who encourage and support their friends and peers to be physically active.
School	% of schools with active school policies (e.g., daily physical education (PE), daily physical activity, recess, “everyone plays” approach, bike racks at school, traffic calming on school property, outdoor time) *. % of schools where the majority (≥80%) of students are taught by a PE specialist *. % of schools where the majority (≥80%) of students are offered the mandated amount of PE (for the given state/territory/region/country) *. % of schools that offer physical activity opportunities (excluding PE) to the majority (>80%) of their students *. % of children and young people who have access to physical activity opportunities at school in addition to PE classes *. % of schools with students who have regular access to facilities and equipment that support physical activity (e.g., gymnasium, outdoor playgrounds, sporting fields, multipurpose space for physical activity, equipment in good condition) *.
Community and Environment	% of children or parents who perceive their community/municipality is doing a good job at promoting physical activity (e.g., variety, location, cost, quality) *. % of communities/municipalities that report they have policies promoting physical activity. % of communities/municipalities that report they have infrastructure (e.g., sidewalks, trails, paths, bike lanes) specifically geared toward promoting physical activity. % of children or parents who report having facilities, programs, parks, and playgrounds available to them in their community *. % of children or parents who report living in a safe neighbourhood where they can be physically active.* % of children or parents who report having well-maintained facilities, parks, and playgrounds in their community that are safe to use.
Government	Evidence of leadership and commitment in providing physical activity opportunities for all children and young people. Allocated funds and resources for the implementation of physical activity promotion strategies and initiatives for all children and young people. Demonstrated progress through the key stages of public policy making (i.e., policy agenda, policy formation, policy implementation, policy evaluation and decisions about the future). HEPA PAT v2 and the scoring rubric published by Ward et al. [20] *
Physical Literacy	% of children and young people who are physically active, physically competent, motivated, confident and possess knowledge and understanding within the cognitive domain of physical literacy *.

* were used in our analysis and those with a strikethrough were changed slightly.

**Table 2 ijerph-19-08138-t002:** The Active Healthy Kids Global Alliance grading rubric.

Grade	Interpretation
A+	94–100%
A	87–93%
A-	80–86%
B+	74–79%
B	67–73%
B−	60–66%
C+	54–59%
C	47–53%
C−	40–46%
D+	33–39%
D	27–33%
D-	20–26%
F	<20%
Inc	Incomplete, insufficient or inadequate information to assign a grade

**Table 3 ijerph-19-08138-t003:** The characteristics of the data sources and the quality indicators these sources used.

Data Source	Sample Size	Age Range	Indicator(s) Used
School Health Research Network Student Health and Wellbeing Survey	110,877	11–16 years	Overall Physical Activity, Sedentary Behaviours, Active Play, Active Transportation, School, Physical Literacy
School Health Research Network School Environment Questionnaire	167 secondary school senior leaders	n/a	School
Health and Attainment of Pupils in a Primary Education Network	1329	8–11 years	Overall Physical Activity, Sedentary Behaviours, Active Transportation, School, Community and Environment, Physical Literacy
School Sport Participation Survey	118,893	7–16 years	Organised Sport Participation, Family and Peers, School, Physical Literacy
School Sport Provision Survey	869 primary school PE teacher/coordinators186 secondary school PE teacher/coordinators	n/a	School
Further Education Sport and Active Lifestyles Survey	3857	16+ years	Organised Sport Participation, Physical Literacy
Play Sufficiency Child Survey Analysis	5884	4–18 years	Active Play, Active Transportation, Family and Peers, Community and Environment
The National Survey for Wales	11,922 1450 950	16+ years 4–11 years 11–16 years	Active Transportation, Family and Peers, Community and Environment
Swan-Linx and Bridge-Linx Data	4778	9–12 years	Physical Fitness
Dragon Challenge	4555	9–12 years	Physical Literacy
Movement Assessment Battery for Children	92	5–7 years	Physical Literacy

**Table 4 ijerph-19-08138-t004:** Grades for all versions of the Active Healthy Kids-Wales Report Card.

Indicator	2014	2016	2018	2021
Overall Physical Activity	D−	D−	D+	F
Organised Sport and Physical Activity	C−	C	C+	C
Active Play	D+	C	C−	C+
Active Transportation	C	C	D+	C−
Sedentary Behaviour	D	D−	F	F
Physical Fitness	N/A	N/A	Inc	C−
Family and Peer Influence	D	D+	D	D+
School	N/A	B	Inc	B−
Community and Environment	B	C	Inc	C
Government	B−	B−	C+	C
Physical Literacy	N/A	Inc	Inc	C−

## Data Availability

The data are available to the research team according to ethical approval. The corresponding author is happy to provide data if required for scrutiny.

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
