# Peer review of "WALES 2021 Active Healthy Kids (AHK) Report Card: The Fourth Pandemic of Childhood Inactivity"

_ijerph, 2022, doi:10.3390/ijerph19138138_

Round 1
Reviewer 1 Report
This article is a report card on children and young people’s physical activity (PA) in Wales. Previous report cards were published in 2014, 2016, and 2018, and this report is based on pre-Covid 19 data. Based on available data and documents (well documented and described in the text), the members of the research working group proposed grades (from F to A+) on 11 different quality indicators of PA (with previously established and used methodology).
The article has a very good structure. The authors wrote a good introduction on the problem of physical (in)activity. The methodology is described in a comprehensive and clear way. The results are clearly presented and Discussion is systematic and based on obtained results. The authors report main findings, and comment on possible shortcomings which may have an effect on the conclusions.
I recommend that this article is published in the present form.
The authors must check the grades in Table 3 – Because there is inconsistency in the Abstract and later in the text, for grades:
- Active play: C+ in the Abstract, and line 203 and C- in the Table 3;
- Active transportation: C- in the Abstract, and line 224 and D+ in the Table 3
The authors should carefully read the paragraph 3.3 (line 203) and 3.4 (line 224) to see if there are any mistakes based on this wrong grades.
Author Response
Thank you for taking the time to review our paper and for your feedback. I have below made a point-by-point response to your suggested edits:
1. I recommend that this article is published in the present form.
Due to the data being collected pre-COVID-19 and the fact that the Report Card is a 2021 Report Card, we feel that this paper is better presented in the past form.
2. The authors must check the grades in Table 3 – Because there is inconsistency in the Abstract and later in the text, for grades.
We have changed the grades for Active Play and Active Transportation throughout the document to ensure that we have reported the correct grade and slightly changed some of the narrative to reflect this mistake. We have also checked all of the other grades.
Reviewer 2 Report
The topic of the paper is interesting therefore there are many problematic aspects
The purpose of the paper is not clear and linked to this difficulty the majority of the problem is focused on the method aspect.
The method are quite confusing and not sufficiently described. The tools are put in a sort of list without clarifying in an adeqate way what of them the authors use.
The sample is not well described and it's not clear what tool and respect to which subjects.
There are also many mistakes and misprint
Author Response
Thank you for taking the time to review our paper and for your feedback. I have below made a point-by-point response to your suggested edits:
1. The purpose of the paper is not clear. We believe that in Line 100/101/102, we clearly state the aim/purpose of the paper.
2. The method are quite confusing and not sufficiently described. The tools are put in a sort of list without clarifying in an adequate way what of them the authors use. We have added more detail to the methods and also included further references to the Active Healthy Kids Global Alliance network, where the methods have been previously described (Line 120 & Line 121).
3. The sample is not well described and it's not clear what tool and respect to which subjects. We have added a sentence (Line 141) about the sample size and age range and that these can be found in Table 3.
4. There are also many mistakes and misprint. We have proofread the manuscript and made changes where there were mistakes and misprint.
Round 2
Reviewer 2 Report
The revision is fine